# Mapping vegetation species succession in a mountainous grassland ecosystem using Landsat, ASTER MI, and Sentinel-2 data

Efosa Gbenga Adagbasa[1]*, Geofrey Mukwada[1,2,3]

1 Department of Geography, University of the Free State, Bloemfontein, South Africa, 2 Afromontane Research Unit, University of the Free State, Bloemfontein, South Africa, 3 Department of Geography, W.A. Franke College of Forestry & Conservation of the University of Montana, Missoula, Montana, United States of America

* Adagbasa.Eg@ufs.ac.za

## Abstract

Vegetation species succession and composition are significant factors determining the rate of ecosystem biodiversity recovery after being disturbed and subsequently vital for sustainable and effective natural resource management and biodiversity. The succession and composition of grasslands ecosystems worldwide have significantly been affected by accelerated environmental changes due to natural and anthropogenic activities. Therefore, understanding spatial data on the succession of grassland vegetation species and communities through mapping and monitoring is essential to gain knowledge on the ecosystem and other ecosystem services. This study used a random forest machine learning classifier on the Google Earth Engine platform to classify grass vegetation species with Landsat 7 ETM+ and ASTER multispectral imager (MI) data resampled with the current Sentinel-2 MSI data to map and estimate the changes in vegetation species succession. The results indicate that ASTER MI has the least accuracy of 72%, Landsat 7 ETM+ 84%, and Sentinel-2 had the highest of 87%. The result also shows that other species had replaced four dominant grass species totaling about 49 km$^2$ throughout the study.

## Introduction

Vegetation succession has many economically significant attributes, including high overall biomass and productivity, a wider variety of species, and minimal nutrients or energy from the ecosystem [1]. Nevertheless, an ecosystem with naturally occurring succession stages will be more resilient to natural and anthropogenic disturbances, suppose these disturbances increase in severity, frequency, and magnitude because of human activities and weather conditions. In that case, the pressure on plant communities increases, causing accelerated succession, creating a new vegetation community, and allowing the succession of non-native species [2].

Luken [1] describes vegetation succession as a change in vegetation composition over 500 years without being disturbed to achieve a climax's stable species composition. Climate change and other disturbances within a short period may result in fluctuations of species composition,

**Data Availability Statement:** The Google earth Engine scripts that contains all the satellite images used for the analysis, random forest classifier and how the results can be exported out of engine has

been cited in the manuscript (https://doi.org/10.6084/m9.figshare.17125043.v2).

**Funding:** The author(s) received no specific funding for this work.

**Competing interests:** The authors have declared that no competing interests exist.

promote non-native species and delay the natural vegetational succession from reaching its climax. The succession of non-native species could impact biodiversity and the natural ecosystem. Non-native invasive species quickly inhabit disturbed spaces and delay native species from achieving seral or climax states. In some cases, the succession is entirely taken over and held for an extended period at an intermediate state, affecting biodiversity [3–5]. Invasive vegetation species threaten native vegetation species and water resources because they grow faster, consume more water, and spread more than the native species [6, 7]. The encroachment of these vegetation species tends to alter the balance of ecosystems, thereby accelerating succession. Vegetation species succession and composition are significant factors determining the rate of ecosystem biodiversity recovery after being disturbed [8] and subsequently vital for sustainable and effective natural resource management and biodiversity. For example, in South Africa, changes in vegetation succession resulting from disturbances have led to significant biodiversity loss [9].

Worldwide, the succession and composition of grasslands ecosystems have been significantly affected by accelerated environmental changes due to natural and anthropogenic activities [10, 11]. It has resulted in shortages in grasslands taxonomy and the efficient functioning of ecosystem services [12]. Grasslands' changing diversity and composition impact ecosystem services like precipitation and temperature controls, freshwater supply, erosion control, and soil formation [13–15]. They can likely result in biodiversity loss [16]. About one-third of South African land surface is covered by the grassland biome [17]. It has less than 3% located in protected areas, and 40–60% have been altered with little chance of being salvaged and returned. It makes the grassland one of the most vulnerable biomes in South Africa [18]. Therefore, understanding spatial data on the succession of grassland vegetation species and communities through mapping and monitoring is essential to gain knowledge on the ecosystem and other ecosystem services [9, 19, 20].

Remote sensing provides an efficient approach for mapping grassland vegetation species by reducing rigorous fieldwork necessitated by standard mapping methods. It does this effectively by offering a wide range of recent data on vegetation species distribution from hyperspectral and multispectral imagery [21, 22]. Extensive studies have been undertaken in monitoring spatio-temporal changes in vegetation species composition and diversity using remote sensing data [23–26]. However, these studies focus briefly on a short period, usually between one to five years, because, before now, only high-resolution hyperspectral images could give accurate vegetation species discrimination at individual levels [9, 19, 27–30]. Osińska-Skotak et al. [24] study the effect of high-resolution hyperspectral ima and LiDAR (Light Detection and Ranging) acquisition date on species identification and, as a result, on classifying individual species in succession trees and shrubs. The researchers also look into the classification accuracy of a particular species is influenced by the research field and the examined environment. The study determined that the time of remote-sensing data collection affects the ability to distinguish succession species. In another study, [31] developed a method for detecting and monitoring the succession process, which is defined as woody vegetation encroachment on non-forest Natura 2000 areas, based on airborne laser scanner (ALS) and hyperspectral (HS) data. Chraibi et al. [32], in their study on changes in tree biodiversity throughout succession, applied both field data and data derived from Sentinel-2 images of 2015 and 2019 to assess variations in tree species richness. They evaluated the benefits and drawbacks of each approach, exploring the potential for remote sensing technology to reveal landscape-level distributions of forest condition and regeneration. Their findings revealed that remote sensing and field data provided distinct insights into tree species compositional changes, as well as alpha- and beta-diversity patterns.

Nevertheless, recent studies have shown that free low-resolution satellite images like Sentinel-2 MSI and Landsat 8 OLI can be used to accurately map and monitor grass vegetation species [33–35]. Vegetation species succession and diversity monitoring can now be done over an extended period using these low-resolution imageries in combination with machine learning (ML).

Because ML can handle nonparametric information with various input predictor data, remote sensing image processing uses ML algorithms [36, 37] to achieve higher accuracy. Increasing accuracy in image classification is a frequent application of machine learning (ML) algorithms, which outperform traditional classifiers for data with many predictor variables [38, 39].

Support Vector Machines (SVM), Classification and Regression Trees (CART), Random Forest (RF), Logistics Regression (LR), Linear Discriminant Analysis (LDA), K-Nearest Neighbour, and Neural Networks are some of the algorithms used in machine learning for image classification. However, numerous studies have shown that random forest performs better in most cases, especially when multispectral images like Sentinel-2 are involved in vegetation species classification [34, 40, 41]. Therefore, this study used a random forest ML algorithm with Landsat 7 ETM+ and ASTER MI data fused with the current Sentinel-2 MSI data to map and estimate the changes in vegetation species succession.

## Study area

The study was conducted at the Golden Gate Highlands National Park, located in Free State, South Africa, near the Lesotho border (Fig 1). The Park covers 340 km$^2$ and is located at the foothills of the Maloti Mountains of the Eastern Free State. The highest peak in the park is 2,829 m (9,281 ft) above sea level. The Park is positioned in the Eastern Highveld region of South Africa and experiences a dry, sunny climate from June to August with showers, hail, and thunderstorms between October and April and snow in winter. The Park has a relatively high annual rainfall of 800 mm (31 in). The park is significant for its rich flora and fauna, which include endemic and endangered plants and animals. The park contains over 60 grass species [42]. The region is a biodiversity hotspot because of the variety of species, including some of the dominant species like Eragrostis, Hyparrhenia cf. Tamba, *T. triandra*, and *M. Capensis* [33, 43].

## Materials and methods

The study used satellite images from different sensors to cover the period of study. The sensors used include the European Union/ESA/Copernicus Sentinel-2 MSI, the United States Geological Survey (USGS) Landsat series ETM+, and the NASA's Land Processes Distributed Active Archive Center (LP DAAC) Advanced Spaceborne Thermal Emission and Reflection Radiometer (ASTER) multispectral imager satellite images with UTM Projection Zone35S, and Datum WGS84 available on the Google earth engine (GEE). The imageries were selected for the rainy months from November to April. The year 2001 was chosen for the Landsat +ETM, 2011 for the ASTER MI, and 2021 for Sentinel-2. Landsat ETM+ and OLI were selected for this study because they have a 15m panchromatic band used to pansharpening the images from 30 m to 15 m. ASTER MI was selected for 2011 because Landsat ETM+ had the scan line error for 2011. Also, the ASTER MI has 15 m resolution bands. The atmospheric and geometric correction was done on the images, and the 10 m bands of Sentinel-2 were then used to resample the pan sharped 15m Landsat images and ASTER MI to 10 m using bicubic interpolation. All the images from the different sensors were calibrated to top-of-atmosphere (TOA) reflectance. ArcGIS 10.7 software was used to generate a hundred random sample points from

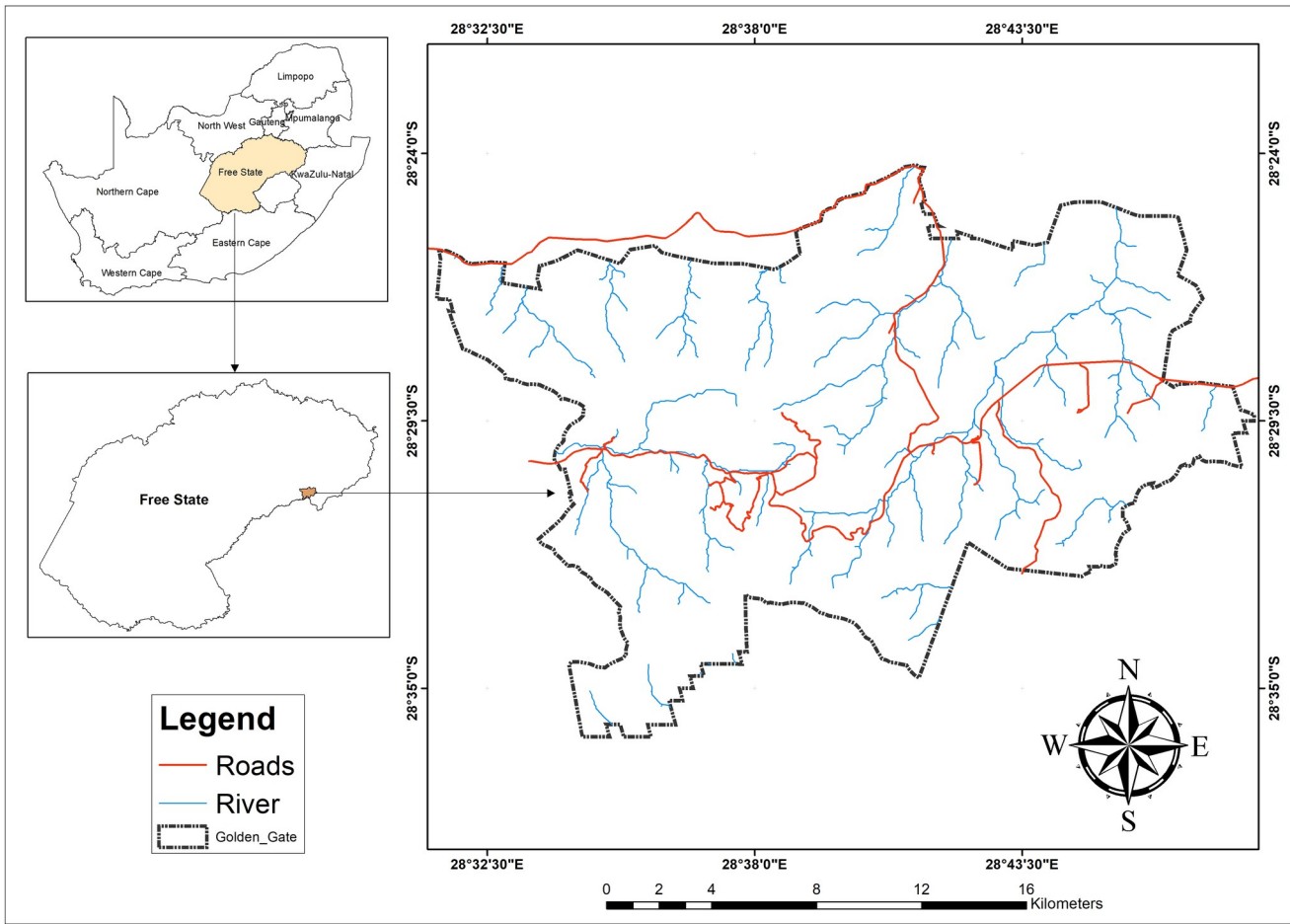

**Fig 1. The study area.**

the spectral signatures of the 12 dominant grass vegetation species in the study area from a previous study [33] that used deep learning and machine learning models to discriminate grass species at the individual level. Their study recommended Sentinel-2 MSI bands 6, 7 (red edge), bands 8 and 8A, band 11, and band 12 to produce optimum classification accuracy. Therefore, the spectral resolution of these bands was used to match and select the bands in the Landsat ETM+ and ASTER MI, as presented in Table 1. The spectral signatures were used for training and cross-validation, with each sample class receiving eight or nine samples to ensure fair representation. The generated samples were saved as a shapefile, imported into GEE, and superimposed on the Landsat ETM+ and ASTER MI and Sentinel-2 image MSI. The sample points were randomly split into a training set (70%) to train the RF classifiers [33, 44] and a test set (30%) for testing purposes [33, 45] in GEE.

The GEE code editor random forest machine learning classifier with ten trees [46] was then used to process the image collections to classify the images into induvial species classes [33, 47]. Ancillary data such as Normalized Difference Vegetation Index (NDVI) were mapped into the image collection on GEE before classification was done to improve classification accuracy. The difference in illuminating effect by high mountains was accounted for by mapping a 15 meters resolution ASTER Global Digital Elevation Model (GDEM) version 2 scale down to 10 m into the image collection [33, 45, 48, 49].

**Table 1. Selected bands for classification.**

| Name | Pixel Size | Wavelength | Description |
|---|---|---|---|
| **Sentinel-2 MSI 2021** | | | |
| B6 | 20 meters | 0.74 μm | Red Edge 2 |
| B7 | 20 meters | 0.78 μm | Red Edge 3 |
| B8 | 10 meters | 0.84 μm | NIR |
| B8A | 20 meters | 0.87 μm | Red Edge 4 |
| B11 | 20 meters | 1.614 μm | SWIR 1 |
| B12 | 20 meters | 2.202 μm | SWIR 2 |
| **ASTER MI 2011** | | | |
| B3N | 15 meters | 0.780–0.860μm | VNIR_Band3N (near infrared, nadir pointing) |
| B04 | 30 meters | 1.600–1.700μm | SWIR_Band4 (short-wave infrared) |
| B05 | 30 meters | 2.145–2.185μm | SWIR_Band5 (short-wave infrared) |
| B06 | 30 meters | 2.185–2.225μm | SWIR_Band6 (short-wave infrared) |
| **Landsat ETM+ 2001** | | | |
| B4 | 30 meters | 0.77–0.90 μm | Near infrared |
| B5 | 30 meters | 1.55–1.75 μm | Shortwave infrared 1 |
| B7 | 30 meters | 2.08–2.35 μm | Shortwave infrared 2 |
| B8 | 15 meters | 0.52–0.90 μm | Panchromatic |

# Results and discussion

## Sensor performance and spectral reflectance

Tables 2–4 shows the accuracy of each sensor. ASTER MI has the least accuracy of 75%, Landsat 7 +ETM 84%, and Sentinel-2 had the highest of 87%. The ASTER MI had the lowest level of accuracy, possibly because [50] stated that each scene does not have all 14 bands. Therefore, some scenes may have fewer bands than others. Hence, only bands 1 to 3 were available for that period. However, these bands' spectral range can be compared to bands 1–5 of Landsat + ETM and OLI. Another possible reason is that the three bands available didn't adequately

**Table 2. Confusion matrix for Sentinel-2 MSI.**

| | EC | PH | SP | AA | SC | TT | AJ | MH | EP | SC | MC | PA | Total | Error of comission |
|---|---|---|---|---|---|---|---|---|---|---|---|---|---|---|
| *E. curvula (EC)* | 5 | 0 | 0 | 0 | 0 | 1 | 0 | 0 | 0 | 0 | 0 | 0 | 6 | 83% |
| *P. Hyparrhenia (PH)* | 0 | 0 | 0 | 0 | 0 | 0 | 0 | 0 | 0 | 0 | 0 | 0 | 0 | 0% |
| *S. Plumosum* | 1 | 0 | 2 | 0 | 0 | 0 | 0 | 0 | 0 | 0 | 0 | 0 | 3 | 67% |
| *A. Asteraceae* | 0 | 0 | 0 | 2 | 0 | 0 | 0 | 0 | 0 | 0 | 0 | 0 | 2 | 100% |
| *S. Centrifugus* | 0 | 0 | 0 | 0 | 2 | 0 | 0 | 0 | 0 | 0 | 0 | 1 | 3 | 67% |
| *T. Triandra* | 0 | 0 | 0 | 0 | 0 | 4 | 0 | 0 | 0 | 0 | 0 | 0 | 4 | 100% |
| *A. Junciformis* | 0 | 0 | 1 | 0 | 0 | 0 | 3 | 0 | 0 | 0 | 0 | 0 | 4 | 75% |
| *M. Hermania* | 0 | 0 | 0 | 0 | 0 | 0 | 0 | 0 | 0 | 0 | 0 | 0 | 0 | 0% |
| *E. Plane Nees* | 0 | 0 | 0 | 0 | 0 | 0 | 0 | 0 | 0 | 0 | 0 | 0 | 0 | 0% |
| *S. Conrathii* | 0 | 0 | 0 | 0 | 0 | 0 | 0 | 0 | 0 | 2 | 0 | 0 | 2 | 100% |
| *M. Capensis* | 0 | 0 | 0 | 0 | 0 | 0 | 0 | 0 | 0 | 0 | 2 | 0 | 2 | 100% |
| *P. Australis* | 0 | 0 | 0 | 0 | 0 | 0 | 0 | 0 | 0 | 0 | 0 | 5 | 5 | |
| **Total** | **6** | **0** | **3** | **2** | **2** | **5** | **3** | **0** | **0** | **2** | **2** | **6** | **31** | |
| **Error of omission** | 83% | 0% | 67% | 100% | 100% | 80% | 100% | 0% | 0% | 100% | 100% | | | |

*Overall accuracy* = (5+0+2+2+2+4+3+0+0+2+2+5) / 31 = 0.87.

0.87 x100 = 87%.

Table 3. Confusion matrix for Landsat 7 ETM+.

| | EC | PH | SP | AA | SC | TT | AJ | MH | EP | SC | MC | PA | Total | Error of comission |
|---|---|---|---|---|---|---|---|---|---|---|---|---|---|---|
| E. curvula (EC) | 3 | 0 | 0 | 0 | 0 | 1 | 0 | 0 | 0 | 0 | 0 | 0 | 4 | 75% |
| P. Hyparrhenia (PH) | 0 | 3 | 0 | 0 | 0 | 0 | 0 | 0 | 0 | 0 | 0 | 1 | 4 | 75% |
| S. Plumosum | 0 | 0 | 0 | 0 | 0 | 0 | 0 | 0 | 0 | 0 | 0 | 0 | 0 | 0% |
| A. Asteraceae | 1 | 0 | 0 | 0 | 0 | 0 | 0 | 0 | 0 | 0 | 0 | 0 | 1 | 0% |
| S. Centrifugus | 0 | 0 | 0 | 0 | 2 | 0 | 0 | 0 | 0 | 0 | 0 | 0 | 2 | 100% |
| T. Triandra | 0 | 0 | 0 | 0 | 0 | 2 | 0 | 0 | 0 | 0 | 0 | 0 | 2 | 100% |
| A. Junciformis | 0 | 0 | 0 | 0 | 0 | 0 | 0 | 0 | 0 | 0 | 0 | 0 | 0 | 0% |
| M. Hermania | 0 | 0 | 0 | 0 | 0 | 0 | 0 | 2 | 0 | 0 | 0 | 0 | 2 | 100% |
| E. Plane Nees | 0 | 0 | 0 | 0 | 0 | 0 | 0 | 0 | 1 | 0 | 0 | 0 | 1 | 100% |
| S. Conrathii | 0 | 0 | 0 | 0 | 0 | 0 | 0 | 0 | 0 | 3 | 0 | 0 | 3 | 100% |
| M. Capensis | 0 | 0 | 0 | 0 | 0 | 1 | 0 | 0 | 0 | 0 | 2 | 0 | 3 | 67% |
| P. Australis | 0 | 0 | 0 | 0 | 0 | 0 | 0 | 0 | 0 | 0 | 0 | 3 | 3 | 100% |
| Total | 4 | 3 | 0 | 0 | 2 | 4 | 0 | 2 | 1 | 3 | 2 | 4 | 25 | |
| Error of omission | 75% | 100% | 0% | 0% | 100% | 50% | 0% | 100% | 100% | 100% | 100% | 75% | | |

*Overall accuracy* = (3+3+0+0+2+2+0+2+1+3+2+3) / 21 = 0.84.

0.84 x100 = 84%.

separate the grass species from each other, as shown in the spectral reflectance curve in Fig 2. Nevertheless, if all the bands were available, ASTER MI should discriminate the grass species effectively to attain a higher accuracy using machine learning classifiers. The accuracy of the ASTER image agrees with a study done by [51]. Their study used ASTER NDVI and EVI to discriminate rice and citrus fields with 75% and 65% accuracy, respectively. They also used Landsat 5 TM NDVI and EVI, which had a lower accuracy of 60% and 65% than the accuracy reached in this study with Landsat 7 ETM+. Landsat 7 ETM+ was able to get a greater accuracy because the bands were pan-sharpened with the 15m panchromatic bands, unavailable on the

Table 4. Confusion matrix for ASTER MI.

| | EC | PH | SP | AA | SC | TT | AJ | MH | EP | Sco | MC | PA | Total | Error of comission |
|---|---|---|---|---|---|---|---|---|---|---|---|---|---|---|
| E. curvula (EC) | 5 | 0 | 0 | 0 | 0 | 0 | 0 | 0 | 0 | 0 | 1 | 1 | 7 | 71% |
| P. Hyparrhenia (PH) | 0 | 1 | 0 | 0 | 0 | 0 | 0 | 0 | 0 | 0 | 0 | 0 | 1 | 100% |
| S. Plumosum (SP) | 0 | 0 | 2 | 0 | 0 | 0 | 0 | 0 | 0 | 0 | 0 | 0 | 2 | 100% |
| A. Asteraceae (AA) | 0 | 0 | 0 | 2 | 0 | 0 | 0 | 0 | 0 | 0 | 0 | 0 | 2 | 100% |
| S. Centrifugus (SC) | 1 | 0 | 0 | 0 | 1 | 0 | 0 | 0 | 0 | 0 | 0 | 0 | 2 | 50% |
| T. Triandra | 1 | 0 | 0 | 0 | 0 | 1 | 0 | 1 | 0 | 0 | 0 | 0 | 3 | 33% |
| A. Junciformis (AJ) | 0 | 0 | 0 | 0 | 0 | 0 | 2 | 0 | 0 | 0 | 0 | 0 | 2 | 100% |
| M. Hermania (MH) | 0 | 0 | 0 | 0 | 0 | 0 | 0 | 0 | 0 | 0 | 0 | 0 | 0 | 0% |
| E. Plane Nees (EP) | 1 | 0 | 0 | 0 | 0 | 0 | 0 | 0 | 0 | 0 | 0 | 0 | 1 | 0% |
| S. Conrathii (Sco) | 0 | 0 | 0 | 0 | 0 | 0 | 0 | 0 | 0 | 1 | 0 | 0 | 1 | 100% |
| M. Capensis (MC) | 0 | 0 | 0 | 0 | 0 | 0 | 0 | 0 | 0 | 0 | 3 | 1 | 4 | 75% |
| P. Australis (PA) | 0 | 0 | 0 | 0 | 0 | 0 | 0 | 0 | 0 | 0 | 0 | 3 | 3 | 100% |
| Total | 8 | 1 | 2 | 2 | 1 | 1 | 2 | 1 | 0 | 1 | 4 | 5 | 26 | |
| Error of omission | 63% | 100% | 100% | 100% | 100% | 100% | 100% | 0% | 0% | 100% | 75% | 60% | | |

*Overall accuracy* = (5+1+2+2+1+1+2+0+0+1+3+3) / 28 = 0.75.

0.75 x100 = 75%.

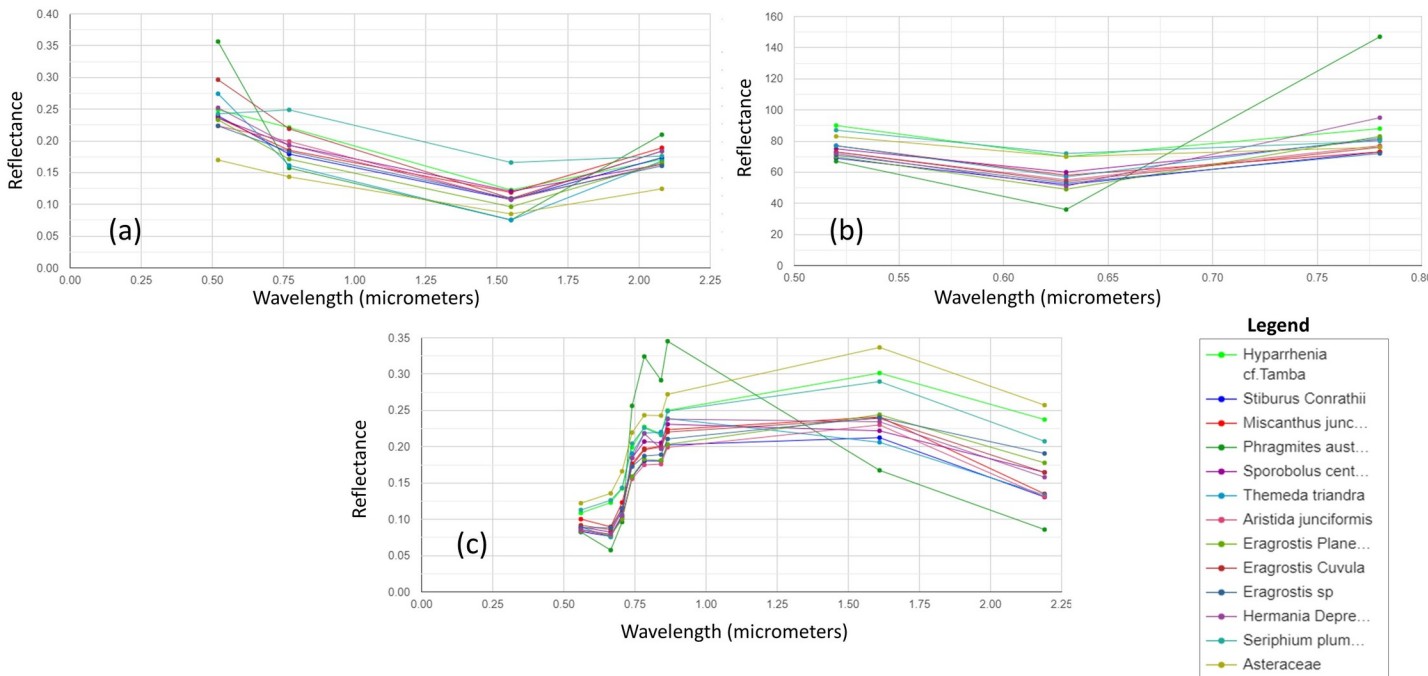

**Fig 2. Species spectral reflectance curves of twelve grass species.** (a) Extracted from Landsat 7 ETM+, (b) Extracted from ASTER MI, (c) Extracted from Sentinel-2.

Landsat 5 TM and resampled from 30 m to10 m using the Sentinel-2 10 m bands. Also, the RF machine learning classifier, which many studies have proved to improve classification [33, 34, 52, 53], contributed to the higher accuracy in this study than the density slicing classification used in their research.

Fig 2 shows the spectral reflectance of the twelve grass species extracted from all the sensors. In the Landsat 7 ETM+, the species were discriminated in wavelengths of 0.52–0.77 μm and 1.55–2.08 μm, representing bands at the start of the wavelength for the panchromatic near-infrared, shortwave infrared 1, and shortwave infrared 2. The ASTER MI has its best spectral separation wavelength of 0.780–0.860μm (VNIR near-infrared, nadir pointing band). At the same time, the Sentinel-2 separated it best in the bands 6, 7 (red edge), bands 8 and 8A, band 11, and band 12 as recommended by the study by [33, 54–56], hence the difference in classification accuracy.

## Grass species changes and succession

Fig 3 shows the map of twelve dominant vegetation species discrimination for 2001, 2011, and 2021. However, the classified map for 2010 was not analyzed further because of the low level of accuracy. The difference of 12% from 2001 and 15% to 2021 might misrepresent the changes that occurred with the classified maps of 2001 and 2021.

Fig 4 shows the vegetation changes from 2001 to 2021. It shows that four grass species had the most significant transformation into other species in area coverage over twenty years. *S. centrifugus* had an enormous shift of 22.6 km$^2$, *E. curvula* had a change of 11.42 km$^2$, *S. Conrathii* had a change of 9.7 km$^2$, and *P.australis* changed 5.14 km$^2$. The other seven grass species had gained and losses over the other four species.

*M. junceus* has the highest success rate. It has replaced different species covering a total land area of 17.22 km$^2$. Another species fast replacing other species is the *T. triandra* species,

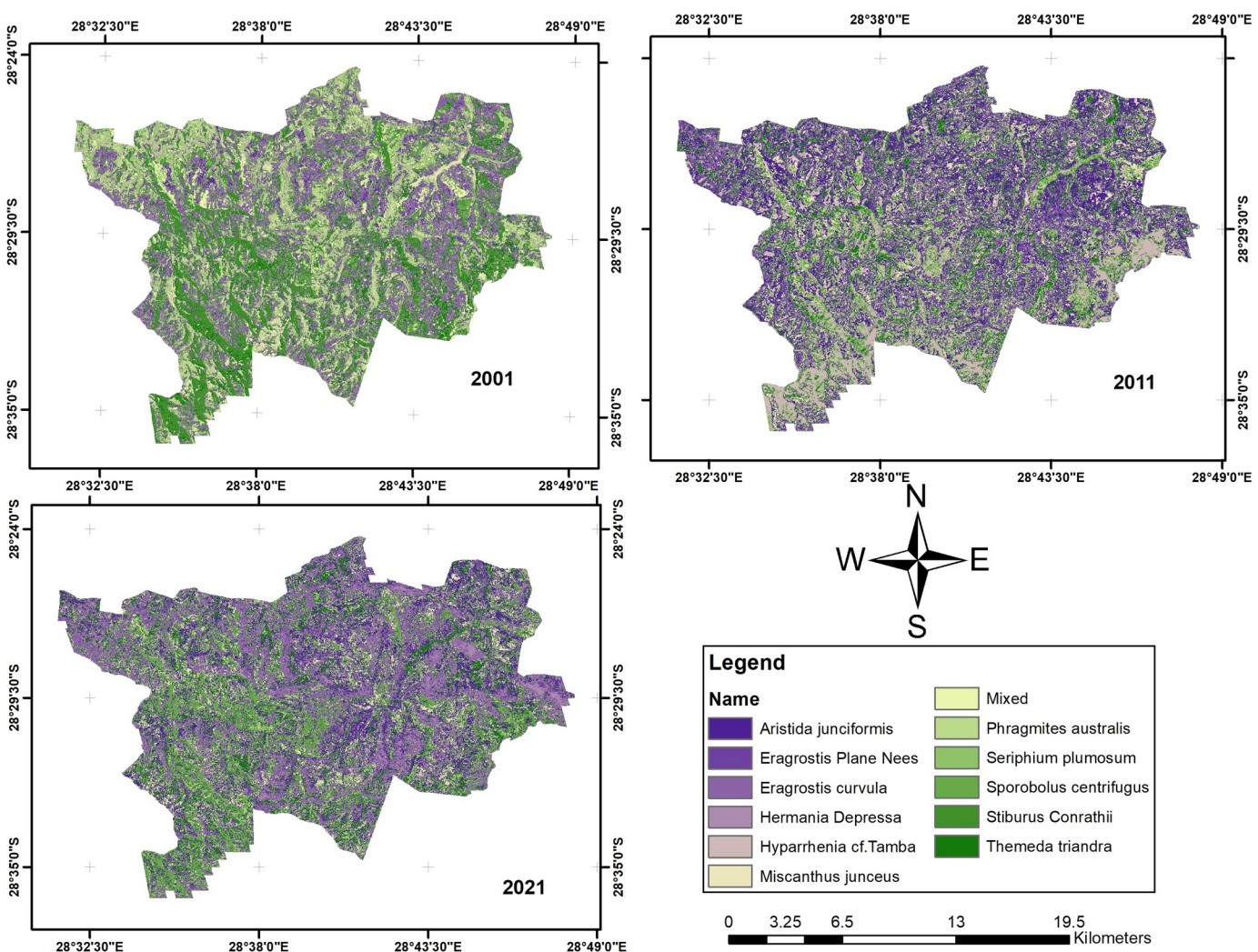

**Fig 3. The map of grass species.** (a) derived from Landsat 7 ETM+, (b) derived from the ASTER MI, (c) derived from Sentinel-2 MSI.

replacing 11.4km$^2$ that formerly contained other species. *E. curvula*, termed an increaser species by many studies (9, 33), and grows very fast in disturbed environments have been replaced by eight different species. *T. triandra* accounts for 50% of the total area that other species have replaced the E. *curvula*. Although the *E. curvula* being an increaser species, it had replaced other species like the *S. centrifugus* and area dominated by mixed species in different study locations and gained back almost 90% of the area lost to other species in Figs 5 and 6. Fig 6 shows that the replacement of E. *curvula* by *T. triandra* happens all over the study area. Still, it is more concentrated around the North, North-East, South-west, and roads of the study area.

*S. centrifugus*, the highest replacement species, is replaced by all the other species in the study area, especially *M. junceus*, *E. curvula*, and *T. triandra*, accounting for 4.8km$^2$, 4.7km$^2$, and 3.3 km$^2$ respectively in Fig 7. The *S.centrifugus* species is monocotyledon and belongs to the *Poaceae family*. It is a native species of South Africa and is termed one of the least concerned threatened species in the red list of South African plants [57]. The succession appears to be occurring around the South, South-western part of the study area, where there are very high elevations (Fig 8).

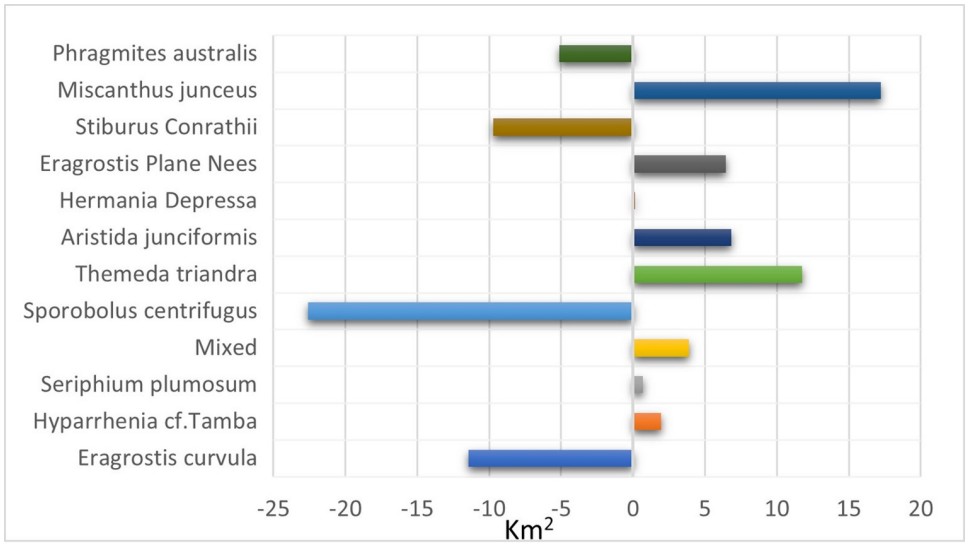

**Fig 4. Vegetation transformation between 2001 and 2021.**

*S. Conrathii* is a native species in South Africa but not endemic to the country. It is also not seen as threatened plant species [57]. This species has about 6.26 km$^2$ replaced by *M. junceus*, majorly in the southwestern (Fig 9) part of the study area but gains 1.06km2 by replacing *S. centrifugus* and 0.34km$^2$ of *H. depressa* (Fig 10).

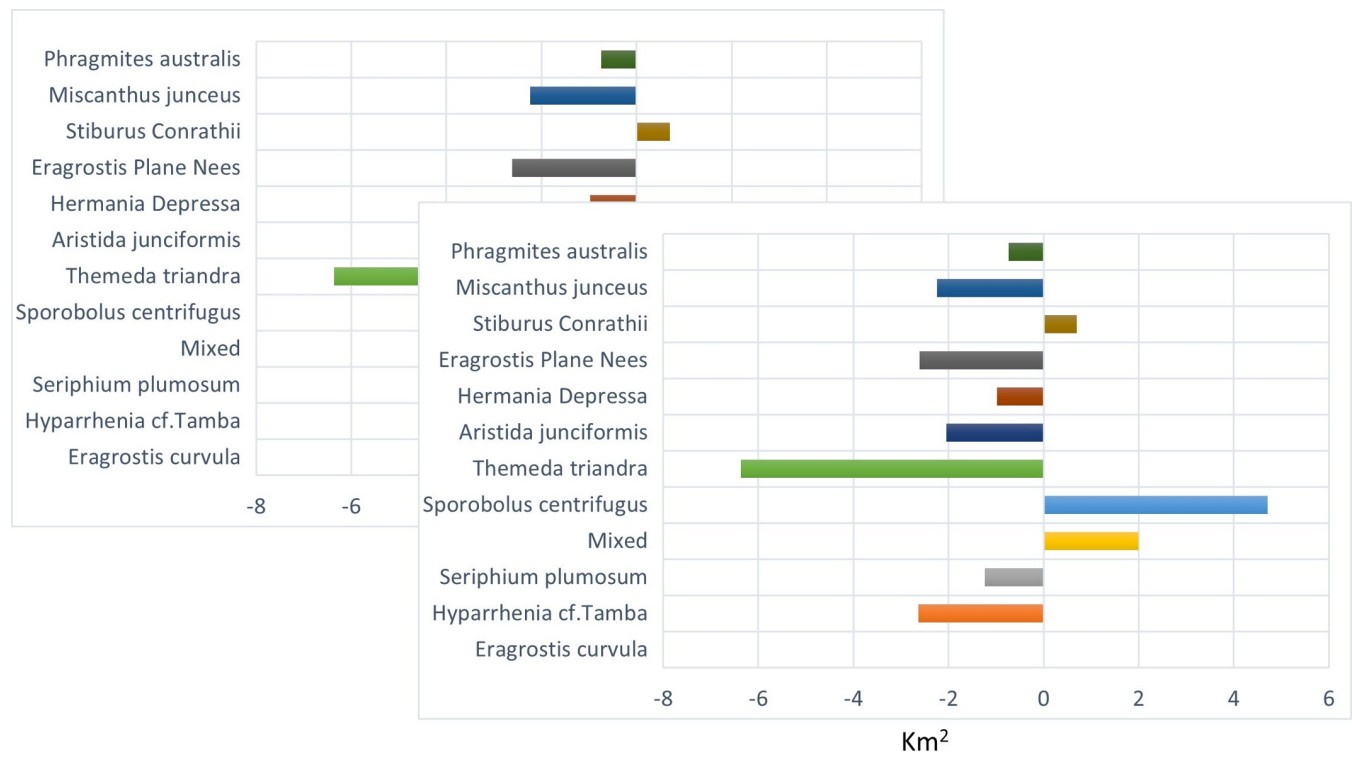

**Fig 5. Species contributions to changes in *Eragrostis curvula*.**

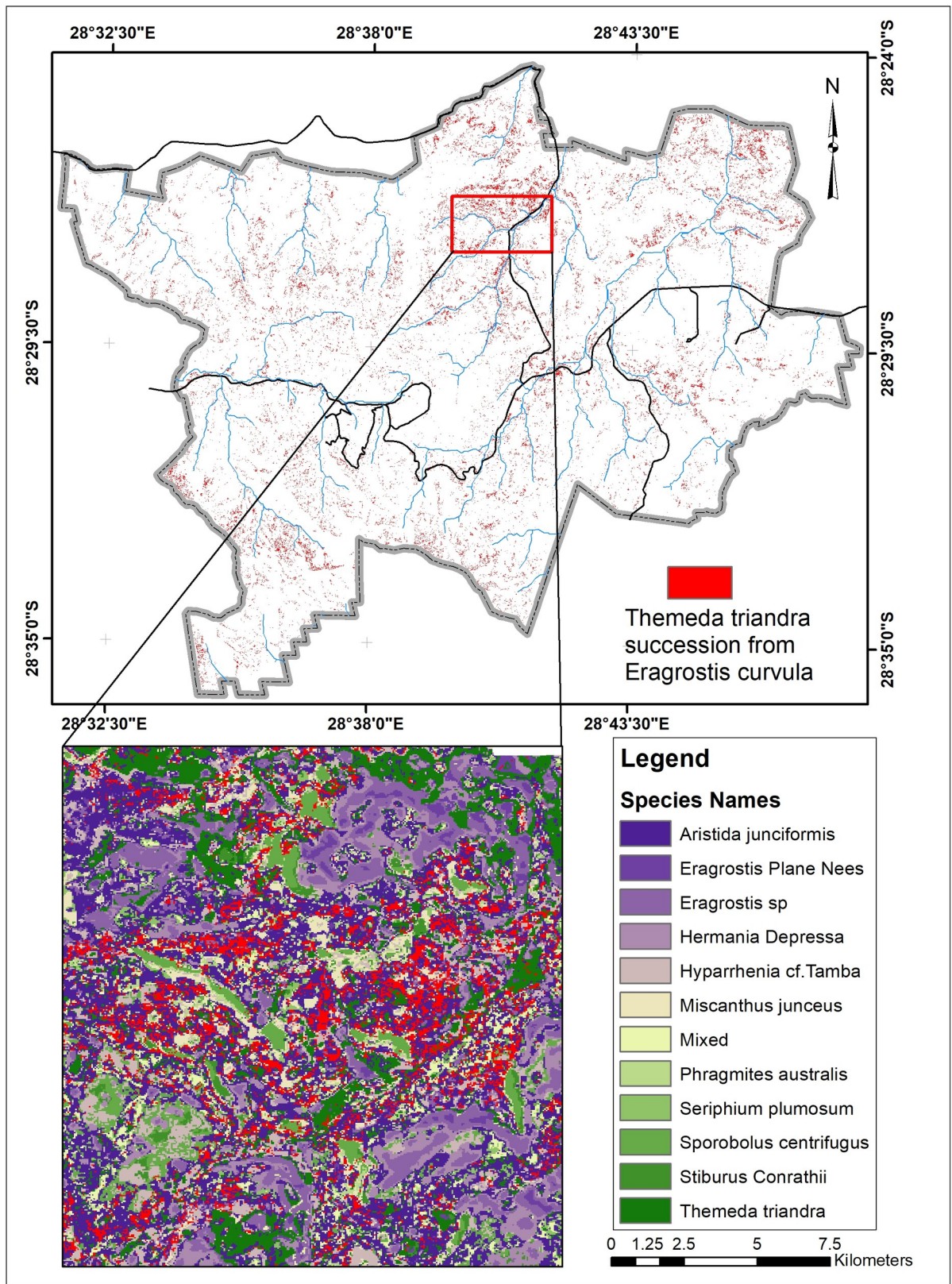

**Fig 6. Areas where *Themeda triandra* succeeded from *Eragrostis curvula*.**

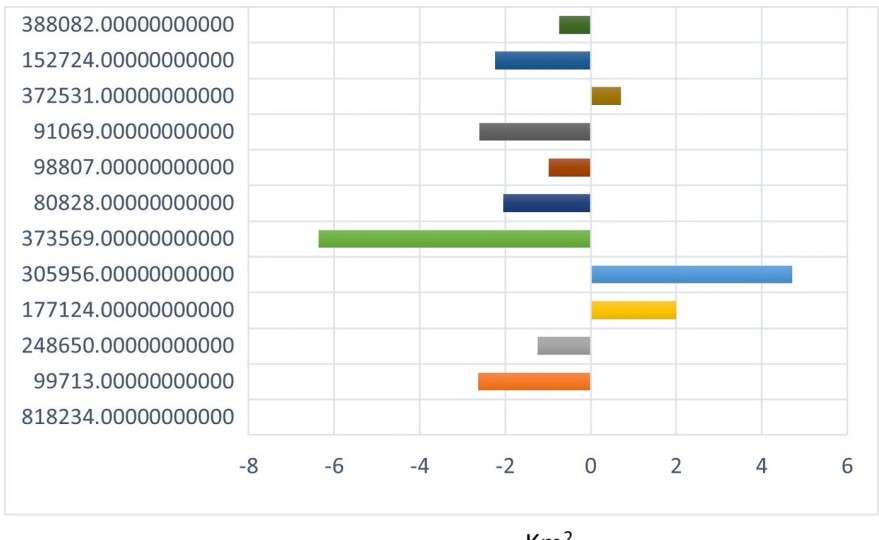

**Fig 7. Species contributions to change in *Sporobolus centrifugus*.**

*P. australis* is a decreaser species quickly affected by overgrazing and has a slow recovery rate after a disturbance [58]. It is a tall grass found across South Africa, especially around river beds and wet environments, and is not at risk of extinction. [57, 59]. Nevertheless, Fig 11 shows that it is being replaced mainly by *T. triandra (2.55 km²), E. plane Nees (2.3 km²), and M. junceus (1.4 km²)*. It is also gaining back by replacing *S. centrifugus* and *E. curvula*. It is found across the study area around the river channels and is replaced mainly in the southern and northern parts of the study area (Fig 12).

The replacement of species may be happening because several factors or disturbances like that could either be natural or anthropogenic. Each species may have a unique or several reasons for replacing others or has been replaced. Some of these species are used for human activities like thatching and medicinal purposes, and some are palatable for grazing. Climate change and fires are common factors that can also affect these successions [60, 61]. The study area is a region constantly affected by wildfires, and the fire severity and magnitude have been mapped by [33]. Their research showed some parts of the study area had constantly been burnt with high fire severity over 20 years. These parts of the study area may be experiencing changes in species composition, leaving only the fire-tolerant species or invasive species like *S. plumosum*, which is a known species that promote the spread of wildfires [62–64]. In fires recently disturbed areas in the park, *S. plumosum* can sprout and lie dormant when encountering higher temperatures and low-moisture conditions for the remainder of winter while awaiting the emergence of spring [65].

Several studies have found that climatic variables like temperature changes significantly impact the distribution of ecological characteristics and environmental dynamics for many types of vegetation species, including alien or native hosts [65–67]. Temperature plays a significant role in the distribution of species, with substantial effects on fire risk. The region's climate often has extended periods of pronounced temperature and low precipitation, resulting in large, devastating fires that devastate populations of plant life [67]. In their research in the study area, Adepoju *et al.* [65] noted the possibility that the distribution of some types of grasses could be better enhanced under conditions with higher daytime temperatures, low to moderate levels of rain at lower elevations. They also noted that climate is an essential factor in

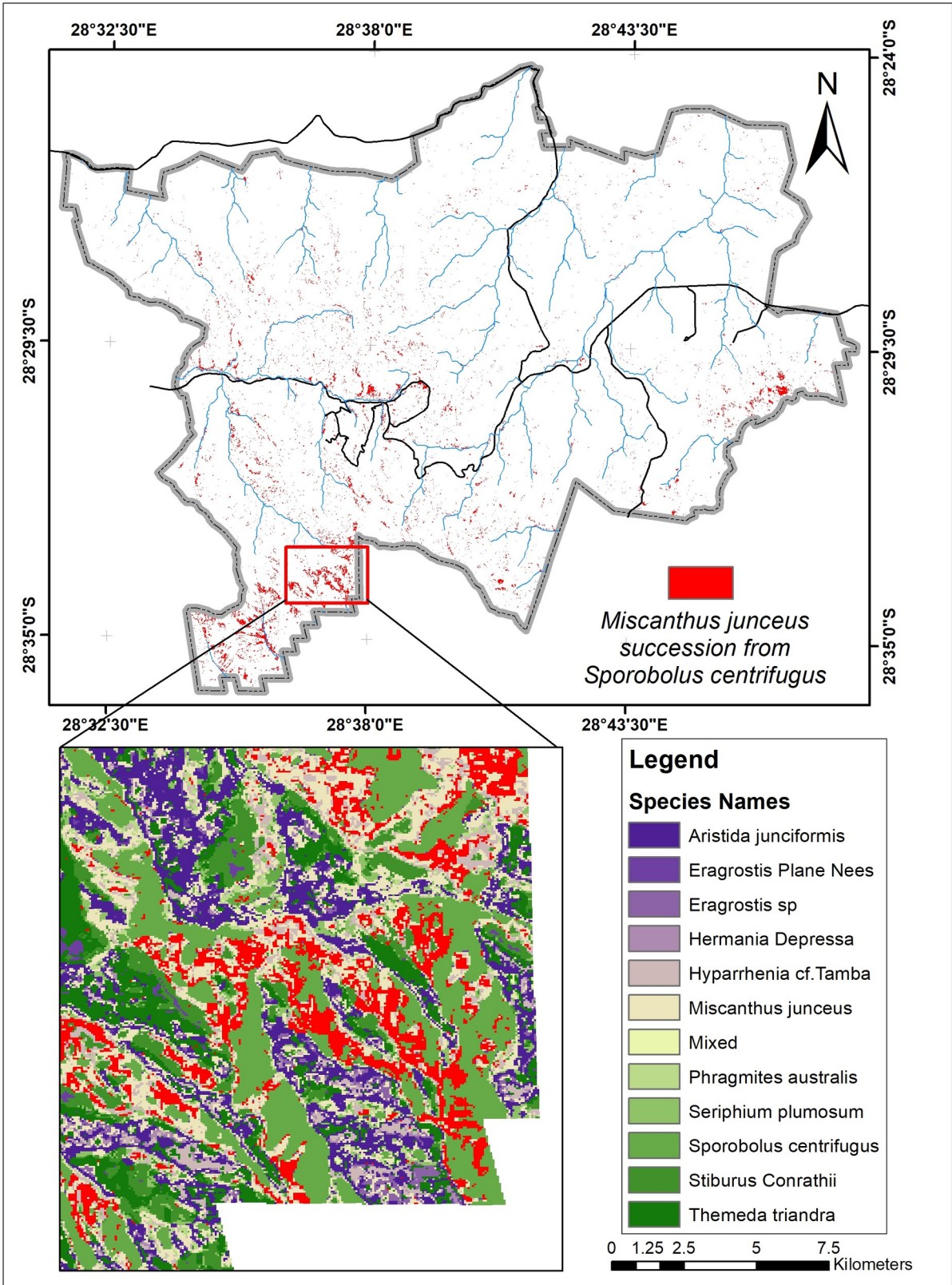

**Fig 8. Areas where *M. junceus* succeeded from *S. centrifugus*.**

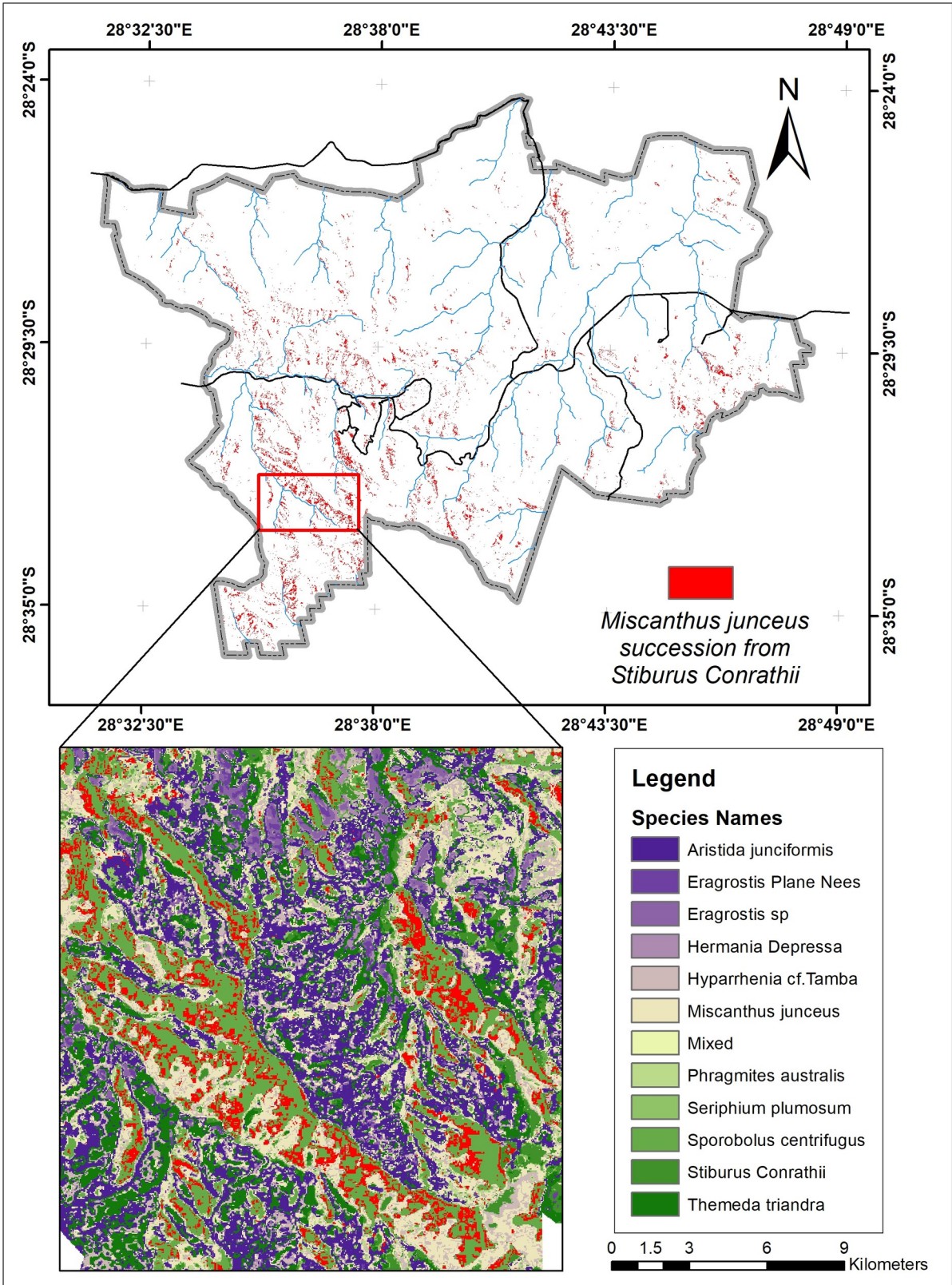

**Fig 9. Areas where *M. junceus* succeeded from *S. Conrathii*.**

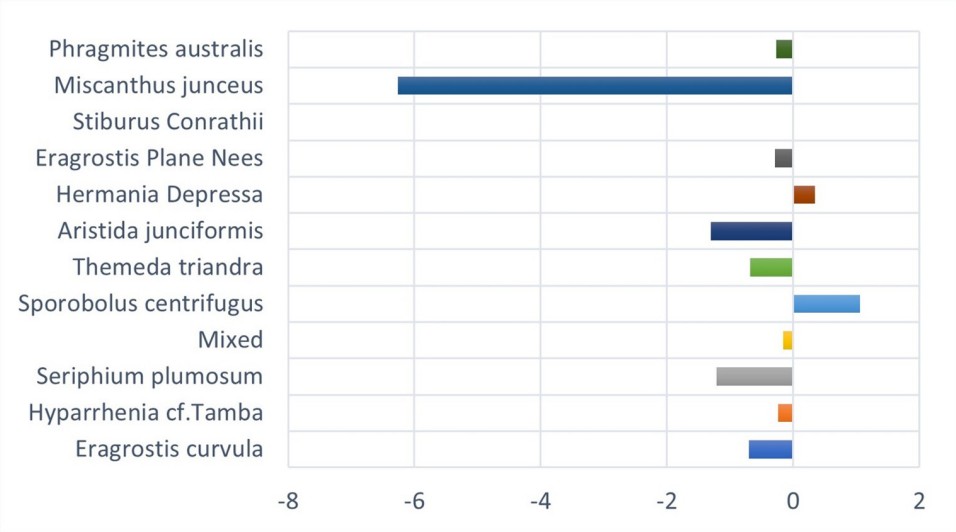

**Fig 10. Species contributions to changes in *Stiburus Conrathii*.**

determining grass species distribution in the mountainous grasslands of South Africa, where periods of warm and cold weather considerably fluctuate.

Areas that are subject to overgrazing and human activities are more likely to experience the spreading of the species that can lead to gaining of new species and loss others. Factors like distance from settlement, land near grasslands and agricultural, and distance from roads impact how disturbances affect different vegetation species [65, 68, 69]. The study area has been a national park with a history of incorporating farms lands to expand the conservation area [70]. In February 1991, the Qwaqwa National Park, which initially comprised multiple crop farmlands, agricultural activities like domestic animal grazing, was also integrated into the study

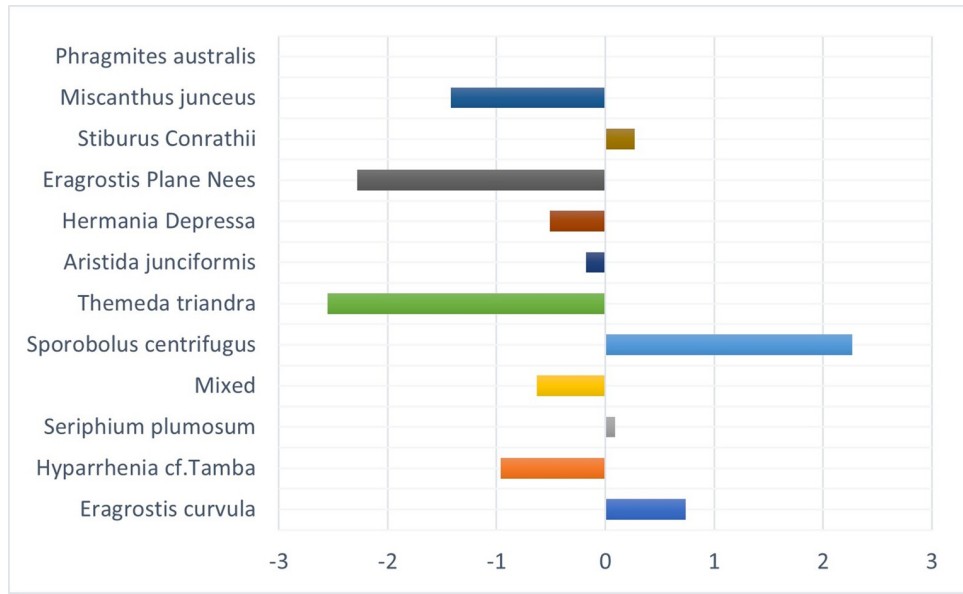

**Fig 11. Species contributions to change in *Phragmites australis*.**

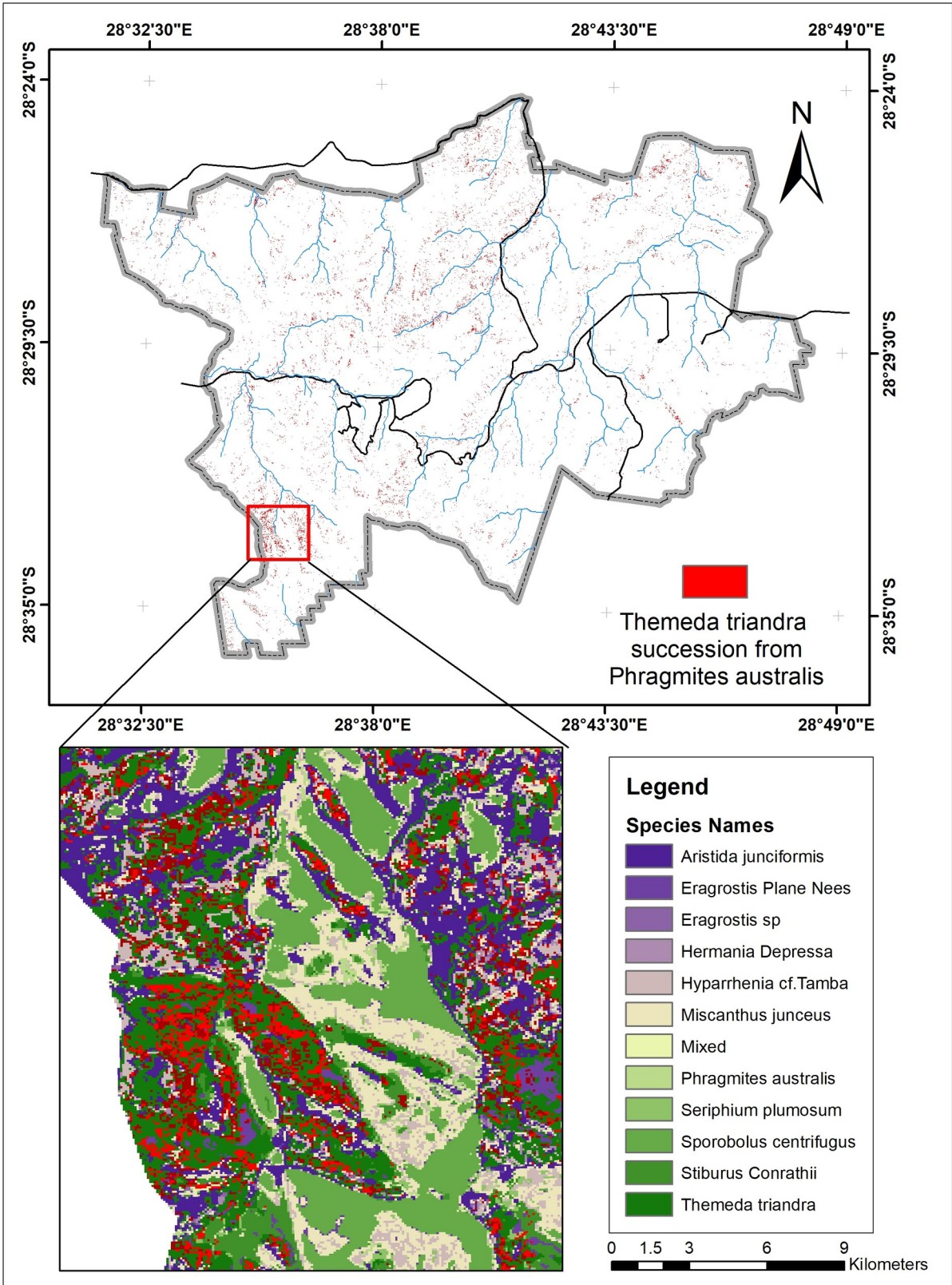

**Fig 12. Areas where *T. triandra* succeeded from *P. australis*.**

area. To date, some of these farmers are still located in the park, and their livestock is still grazing the vegetation [70, 71]. However, some farmers have stopped planting on them. The disturbed land has been left to recover by the park managers as a conservation strategy. Nevertheless, not all these previous farmlands have recovered fully because the soils have been over-exploited. The vegetation species in these locations struggle to survive with the effects of disturbances like frequent wildfires and overgrazing from agricultural animals from within the park or the surrounding communal areas or from the herbivorous animals within the park.

## Conclusion

This study has shown that the Landsat 7 ETM+ can be used for vegetation species discrimination if the panchromatic band is used to pan sharpening the 30 m bands to 15 m and then resampled with the 10m bands Sentinel-2 MSI. It will allow for research in monitoring vegetation species changes over a long period. Although the ASTER MI wasn't used to analyze the vegetation species changes, it also has a prospect of being used if all the recommended bands are available. The study also confirmed many other studies: using ML techniques such as RF with freely accessible Sentinel-2 MSI can identify grass species with high accuracy. The study explored the differences in vegetation species that have occurred over 20 years but didn't explore the precise reasons why others were replacing some vegetation species. The causes and factors influencing the shift in vegetation species in some park locations can be done in a further study. It will help the park managers appropriately manage the park and prevent key vegetation species from total annihilation by other more aggressive vegetation species, preserving the animal population in the park and keeping the ecosystem healthy.

## Acknowledgments

The authors would like to acknowledge the support of the South African National Park (SAN-PARKS) South Africa throughout data collection. The authors would also like to acknowledge the contribution of Prof. S.A Adelabu without whom this research would not have been possible.

## Author Contributions

**Conceptualization:** Efosa Gbenga Adagbasa.

**Data curation:** Efosa Gbenga Adagbasa.

**Formal analysis:** Efosa Gbenga Adagbasa.

**Investigation:** Efosa Gbenga Adagbasa.

**Methodology:** Efosa Gbenga Adagbasa.

**Resources:** Efosa Gbenga Adagbasa.

**Supervision:** Geofrey Mukwada.

**Validation:** Efosa Gbenga Adagbasa.

**Writing – original draft:** Efosa Gbenga Adagbasa.

**Writing – review & editing:** Efosa Gbenga Adagbasa, Geofrey Mukwada.

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
