## [Decision Letter · Decision Letter 0]

17 Sep 2021

PONE-D-21-25779Mapping Vegetation Species Succession in a Mountainous Grassland ecosystem using Landsat and Sentinel-2 dataPLOS ONE

Dear Dr. Adagbasa,

Thank you for submitting your manuscript to PLOS ONE. After careful consideration, we feel that it has merit but does not fully meet PLOS ONE’s publication criteria as it currently stands. Therefore, we invite you to submit a revised version of the manuscript that addresses the points raised during the review process.

We look forward to receiving your revised manuscript.

Kind regards,

Bijeesh Kozhikkodan Veettil

Academic Editor

PLOS ONE

Journal Requirements:

2. We note that Figure 1, 3, 6, 8, 10 and 12 in your submission contain map/satellite images which may be copyrighted. All PLOS content is published under the Creative Commons Attribution License (CC BY 4.0), which means that the manuscript, images, and Supporting Information files will be freely available online, and any third party is permitted to access, download, copy, distribute, and use these materials in any way, even commercially, with proper attribution. For these reasons, we cannot publish previously copyrighted maps or satellite images created using proprietary data, such as Google software (Google Maps, Street View, and Earth). For more information, see our copyright guidelines: http://journals.plos.org/plosone/s/licenses-and-copyright.

a) You may seek permission from the original copyright holder of Figures 1, 3, 6, 8, 10 and 12  to publish the content specifically under the CC BY 4.0 license.  

3. We noted in your submission details that a portion of your manuscript may have been presented or published elsewhere. [Data on grass species spectral signatures for image classification and Figure 1 (study area map) were taken from one of my published manuscripts. Including this data does not constitute dual publication because this manuscript focuses on developing a section of the previous research by testing and using a different methodology.] Please clarify whether this publication was peer-reviewed and formally published. If this work was previously peer-reviewed and published, in the cover letter please provide the reason that this work does not constitute dual publication and should be included in the current manuscript.

Reviewers' comments:

Reviewer's Responses to Questions

**Comments to the Author**

1. Is the manuscript technically sound, and do the data support the conclusions?

Reviewer #1: Partly

Reviewer #2: Yes

2. Has the statistical analysis been performed appropriately and rigorously? 

Reviewer #1: Yes

Reviewer #2: Yes

3. Have the authors made all data underlying the findings in their manuscript fully available?

Reviewer #1: Yes

Reviewer #2: Yes

4. Is the manuscript presented in an intelligible fashion and written in standard English?

Reviewer #1: Yes

Reviewer #2: Yes

5. Review Comments to the Author

Reviewer #1: Title of the Manuscript:

Mapping Vegetation Species Succession in a Mountainous Grassland ecosystem using Landsat and Sentinel-2 data

General Remarks:

A.) The studies related to vegetation species succession are highly useful scientific sector to understand the impact of climate change on ecosystems. However, the main hurdle in this application is that the species succession studies are highly long term studies. Recent advancements in remote sensing data acquisitions yielded high resolution data through with mapping the species is now no more tedious. But the challenge in understanding the species succession is non-availability of high resolution imagery for historic dates. To overcome this challenge, authors have used machine learning techniques.

B.) At the outset, authors should first tell why they used machine learning techniques to do the change detection – viz-a-viz, species identification from the multi-date and multi-platform satellite data. What inspired/motivated and need aspect of using machine learning is to be specified.

C.) Authors have used only Random Forest classifier. Is there any other ML technique available? If so, should be addressed.

D.) Authors failed to explain the reasons for replacement of species – however they have simply attributed the reasons to several factors or disturbances (natural or anthropogenic) – This is completely putting vagueness in the article. The enthusiasm kept in the introduction has been lost due to this. Otherwise this article could have contributed value to the scientific domain

Other Remarks:

1.) Abstract:

a. “The results indicate that ASTER IM has the…” � IM? Also, many times in the manuscript this needs correction (from ASTER IM to ASTER MI)

2. If ASTER data is used then why it is not in the title of the article?

b.) Introduction Section

a. Authors could give some more information about similar work carried out by other researchers

3.) Study Area

a. Authors could elaborate more information about dominant (top 5) grass species (however it was specified as 60 grass species are available.)

4.) Materials and Methods

a. Authors could elaborate more information about dominant (top 5) grass species (however it was specified as 60 grass species are available)

5.) Results and Discussion

a. Authors have abruptly mentioned the accuracy of each sensor – they could have mentioned the process/parameters for deriving the accuracy of each sensor

b. Authors should mention the reason for replacement/or species succession for each species independently and categorically – this will add more value to the article

6.) Conclusion

a. Now it is known that Sentinel-2 MSI is more superior to other available space-borne sensors. The conclusion section is completely missing the mentioning about machine learning algorithms and its applications in this technical area.

Reviewer #2: This work used Landsat 7 EMT+, ASTER MI and Sentinel-2 MSI data to map and estimate the changes in vegetation species succession with accuracies more than 70%. This is helpful for understanding the spatial data on the succession of grassland vegetation species and communities and gain knowledge on the ecosystem and ecosystem services South Africa with large areas of grassland. As the authors stated that high-resolution hyperspectral images can give accurate vegetation species, I’d like to see how the authors obtain the 12 types of vegetation species by using the low spectral resolution images from ETM+, ASTER and Sentinel-2, unfortunately, I did not find the way the authors worked out in the section of materials and methods. I suggest the authors should explain the method more clearly and the whole procedures to classify the image into such detailed vegetation such that the readers can understand your work well. My major concerns are the below:

Where are the locations of the 12 vegetation species samples did you collect, and how many sites did you collect?

What are the major differences between these vegetation species and what typical features did you use to input into the classifier?

6. PLOS authors have the option to publish the peer review history of their article (what does this mean?). If published, this will include your full peer review and any attached files.

Reviewer #1: No

Reviewer #2: No

---

## [Author Response · Author response to Decision Letter 0]

4 Nov 2021

Response to Review Comments

Journal Requirements 

Response: The manuscript meets all PLOS ONE's style requirements

2. We note that Figure 1, 3, 6, 8, 10 and 12 in your submission contain map/satellite images which may be copyrighted. All PLOS content is published under the Creative Commons Attribution License (CC BY 4.0), which means that the manuscript, images, and Supporting Information files will be freely available online, and any third party is permitted to access, download, copy, distribute, and use these materials in any way, even commercially, with proper attribution. For these reasons, we cannot publish previously copyrighted maps or satellite images created using proprietary data, such as Google software (Google Maps, Street View, and Earth).

Response: Figure 1 was created by me and was used in a previously peer-reviewed and published work. Nevertheless, I have redesigned another study area map to replace figure 1. Figures 3, 6, 8, 10, and 12 are maps showing different analysis results for this manuscript. The analysis was carried out from data generated from field studies and freely available satellite images, e.g., Landsat, ASTER, and Sentinel II. These images have no restrictions on reuse, sale, or redistribution. They only require that the author include a statement of the data source in their manuscripts. Nevertheless, figures 3, 6, 8, 10, and 12 were also redesigned using a different color symbolization.

3. We noted in your submission details that a portion of your manuscript may have been presented or published elsewhere. [Data on grass species spectral signatures for image classification and Figure 1 (study area map) were taken from one of my published manuscripts. Including this data does not constitute dual publication because this manuscript focuses on developing a section of the previous research by testing and using a different methodology.] Please clarify whether this publication was peer-reviewed and formally published. If this work was previously peer-reviewed and published, in the cover letter please provide the reason that this work does not constitute dual publication and should be included in the current manuscript.

Response: Figure 1 was created by me and was used in a previously peer-reviewed and published work. Nevertheless, I have redesigned another study area map to replace figure 1. The reason that the data on grass species spectral signatures for image classification does not constitute dual publication has been included in the cover letter.

4. We note that you have stated that you will provide repository information for your data at acceptance. Should your manuscript be accepted for publication, we will hold it until you provide the relevant accession numbers or DOIs necessary to access your data. If you wish to make changes to your Data Availability statement, please describe these changes in your cover letter, and we will update your Data Availability statement to reflect the information you provide.

Response: Changes to the data availability statement have been described in the cover letter.

Response: the abstract in the manuscript and online submission have been amended and are the same.

Specific Response to Reviewer Comments

Reviewer: 1

Comments to the Author

General Remarks:

A.) The studies related to vegetation species succession are highly useful scientific sector to understand the impact of climate change on ecosystems. However, the main hurdle in this application is that the species succession studies are highly long term studies. Recent advancements in remote sensing data acquisitions yielded high resolution data through with mapping the species is now no more tedious. But the challenge in understanding the species succession is non-availability of high resolution imagery for historic dates. To overcome this challenge, authors have used machine learning techniques.

B.) At the outset, authors should first tell why they used machine learning techniques to do the change detection – viz-a-viz, species identification from the multi-date and multi-platform satellite data. What inspired/motivated and need aspect of using machine learning is to be specified.

Response: The reasons and motivation for using the machine learning method has been added in the manuscript (line 83-92)

C.) Authors have used only Random Forest classifier. Is there any other ML technique available? If so, should be addressed.

 Response: The reasons why the only random forest was used above other classifier has been included. Also, other ML techniques have been mentioned (Line 90-92)

D.) Authors failed to explain the reasons for the replacement of species – however, they have simply attributed the reasons to several factors or disturbances (natural or anthropogenic) – This is completely putting vagueness in the article. The enthusiasm kept in the introduction has been lost due to this. Otherwise, this article could have contributed value to the scientific domain.

Response: The factors weren't included because they weren't the focus of the study. Authors only give possible reasons for vegetation succession from literature in the introduction and streamline the focus of the study on accurately understanding spatial data on the succession of grassland vegetation species and communities through mapping and monitoring. Nevertheless, the authors are already working on more detailed research on the reasons for species replacement.

Other Remarks:

1.) Abstract:

a. "The results indicate that ASTER IM has the…" � IM? Also, many times in the manuscript this needs correction (from ASTER IM to ASTER MI)

Response: All ASTER IM have been corrected to ASTER MI

2. If ASTER data is used then why it is not in the title of the article?

Response: ASTER MI has been added to the title.

b.) Introduction Section

a. Authors could give some more information about similar work carried out by other researchers

Response: similar work carried out by other researchers was mentioned in the introduction. Nevertheless, more details of there research have been added to the introduction. 

3.) Study Area

a. Authors could elaborate more information about dominant (top 5) grass species (however it was specified as 60 grass species are available.)

Response: The top four dominat spcies have been added to the study area section

4.) Materials and Methods

a. Authors could elaborate more information about dominant (top 5) grass species (however it was specified as 60 grass species are available)

Response: The authors wrote that the spectral library of 12 dominant species was obtained from a previous study and referred to it. The names of these dominant species may be found in the aforementioned paper.

5.) Results and Discussion

a. Authors have abruptly mentioned the accuracy of each sensor – they could have mentioned the process/parameters for deriving the accuracy of each sensor

Response: The erorr matrix and xplaination have been added to the result section 

b. Authors should mention the reason for replacement/or species succession for each species independently and categorically – this will add more value to the article

Response: The reason weren't included because they weren't the focus of the study. However, authors give possible reasons for vegetation succession from literature Nevertheless, the authors are already working on a more detailed research on the reasons for species replacement which will be the focus of another article.

6.) Conclusion

a. Now it is known that Sentinel-2 MSI is more superior to other available space-borne sensors. The conclusion section is completely missing the mentioning about machine learning algorithms and its applications in this technical area.

Response: Although the research introduction has already established that Sentinel-2 MSI is more superior to other available space-borne sensors, It has also been added to the conclusion section. 

Reviewer: 2

Comments to the Author

This work used Landsat 7 EMT+, ASTER MI and Sentinel-2 MSI data to map and estimate the changes in vegetation species succession with accuracies more than 70%. This is helpful for understanding the spatial data on the succession of grassland vegetation species and communities and gain knowledge on the ecosystem and ecosystem services South Africa with large areas of grassland. As the authors stated that high-resolution hyperspectral images can give accurate vegetation species, I'd like to see how the authors obtain the 12 types of vegetation species by using the low spectral resolution images from ETM+, ASTER and Sentinel-2, unfortunately, I did not find the way the authors worked out in the section of materials and methods. I suggest the authors should explain the method more clearly and the whole procedures to classify the image into such detailed vegetation such that the readers can understand your work well. My major concerns are the below:

Where are the locations of the 12 vegetation species samples did you collect, and how many sites did you collect?

What are the major differences between these vegetation species and what typical features did you use to input into the classifier?

 Response: According to the methodology sections, the spectral signatures of the 12 dominant species were taken from previous research and referred to it. The necessary information on how the fieldwork, image processing, and spectral signature development were documented in that study. Because repeating it would appear like research duplication, the authors didn't want to bring it up again.

In the methodology section, the authors have attempted to make image processing more evident. The locations and number of each vegetation species were determined in a previous study mentioned above. This research produced 100 random samples by combining digital numbers representing spectral data from the results of the previous study with spectral information from images to generate 100 random samples.

The differences between these vegetation species are shown in the spectral reflectance curve in figure 2. The typical features used to input into the classifier was a number of 10 trees, the bands that can easily separate the species (see Figure 2 and Table 1), NDVI, and elevation data. The authors stated all these in the methodology section.

---

## [Decision Letter · Decision Letter 1]

26 Nov 2021

Mapping vegetation species succession in a mountainous grassland ecosystem using Landsat, ASTER MI, and Sentinel-2 data

PONE-D-21-25779R1

Dear Dr. Adagbasa,

We’re pleased to inform you that your manuscript has been judged scientifically suitable for publication and will be formally accepted for publication once it meets all outstanding technical requirements.

Kind regards,

Bijeesh Kozhikkodan Veettil

Academic Editor

PLOS ONE

Additional Editor Comments (optional):

Reviewers' comments:

Reviewer's Responses to Questions

**Comments to the Author**

1. If the authors have adequately addressed your comments raised in a previous round of review and you feel that this manuscript is now acceptable for publication, you may indicate that here to bypass the “Comments to the Author” section, enter your conflict of interest statement in the “Confidential to Editor” section, and submit your "Accept" recommendation.

Reviewer #1: All comments have been addressed

Reviewer #2: All comments have been addressed

2. Is the manuscript technically sound, and do the data support the conclusions?

Reviewer #1: Yes

Reviewer #2: Yes

3. Has the statistical analysis been performed appropriately and rigorously? 

Reviewer #1: Yes

Reviewer #2: Yes

4. Have the authors made all data underlying the findings in their manuscript fully available?

Reviewer #1: Yes

Reviewer #2: (No Response)

5. Is the manuscript presented in an intelligible fashion and written in standard English?

Reviewer #1: Yes

Reviewer #2: (No Response)

6. Review Comments to the Author

Reviewer #1: (No Response)

Reviewer #2: Thanks authors' hard work in addressing the problems I concerned. I have no other concerns in the revised version of the manuscript.

7. PLOS authors have the option to publish the peer review history of their article (what does this mean?). If published, this will include your full peer review and any attached files.

Reviewer #1: No

Reviewer #2: No

---

## [Editor Report · Acceptance letter]

12 Jan 2022

PONE-D-21-25779R1 

Mapping vegetation species succession in a mountainous grassland ecosystem using Landsat, ASTER MI, and Sentinel-2 data 

Dear Dr. Adagbasa:

I'm pleased to inform you that your manuscript has been deemed suitable for publication in PLOS ONE. Congratulations! Your manuscript is now with our production department. 

Kind regards, 

on behalf of

Dr. Bijeesh Kozhikkodan Veettil 

Academic Editor

PLOS ONE